# Preparation of Poly(vinyl Alcohol) Microparticles for Freeze Protection of Sensitive Fruit Crops

**DOI:** 10.3390/polym14122452

**Published:** 2022-06-16

**Authors:** Constanza Sabando, Walther Ide, Saddys Rodríguez-Llamazares, Richard M. Bastías, Miguel Valenzuela, Claudio Rojas, Johanna Castaño, Natalia Pettinelli, Rebeca Bouza, Niels Müller

**Affiliations:** 1Centro de Investigación de Polímeros Avanzados, Avenida Collao 1202, Edificio Laboratorio CIPA, Concepción 4051381, Chile; c.sabando@cipachile.cl (C.S.); w.ide@cipachile.cl (W.I.); c.rojas@cipachile.cl (C.R.); johanna.castano@uss.cl (J.C.); n.pettinelli@cipachile.cl (N.P.); rebeca.bouza@udc.es (R.B.); 2Departamento de Producción Vegetal, Universidad de Concepción, Av. Vicente Méndez 595, Chillán 3780000, Chile; ribastias@udec.cl (R.M.B.); mivalenzuela@udec.cl (M.V.); 3Facultad de Ingeniería y Tecnología, Universidad San Sebastián, Lientur 1457, Concepción 4080871, Chile; 4Grupo de Polímeros, Departamento de Física y Ciencias de la Tierra, Escuela Universitaria Politécnica, Universidad de A Coruña, Serantes, Avda. 19 de Febrero s/n, 15471 Ferrol, Spain; 5Unidad de Desarrollo Tecnológico, Universidad de Concepción, Avenida Cordillera 2634, Parque Industrial Coronel, Coronel 4191996, Chile; n.muller@udt.cl

**Keywords:** PVA microparticles, ice recrystallization inhibition, freeze injury, cherry plants

## Abstract

Poly(vinyl alcohol) (PVA) displays ice recrystallization inhibition (IRI) properties as many antifreeze proteins found in cold tolerant organisms. The molecular architecture and composition (molecular weight and distribution of pendant OH and acetate groups) have been studied to improve the antifreezing properties of PVA, suggesting that the molecular architecture of PVA plays an important role in IRI activity. The present work deals with the preparation of PVA microparticles using an alkaline treatment. The effect of PVA molecular weight on the morphology and antifreezeing properties of PVA microparticles was investigated. The antifreezeing property of PVA microparticles on the susceptibility of flower bud tissues to freeze damage was also evaluated. The alkaline treatment of an aqueous PVA solution produced stable polymer chain aggregates with spherical shapes. The average size of the PVA microparticles increased significantly with the increasing molecular weight of the PVA macromolecule precursor. The PVA microparticles inhibited the growth of ice crystals and blocked ice growth at concentrations as low as 0.01 % *w*/*v*. The effect of impeding ice crystal growth by preventing the joining of adjacent ice crystals is attributed to the larger size of the PVA particles adsorbed on the ice surface compared to the aggregated PVA macromolecules in saline solution. The thermal hysteresis activity of PVA macromolecules and microparticles was not detected by differential scanning calorimetry analysis. The PVA microparticles reduced the incidence of freeze injuries in flower bud tissues by 55% and their application, considering the low toxicity of PVA, has a high potential for freeze protection in fruit crops.

## 1. Introduction

Late spring frosts are one of the most damaging weather phenomena for fruit crops, causing severe economic losses to the fruit industry worldwide each year. Zohner et al. [1] analyzed the occurrences of late spring frosts between 1959 and 2017 in Europe, East Asia, and North America, and found that this damaging weather phenomenon increased by 70 and 72% in the biome area of Europe and East Asia, respectively, and by about 50% in North America. In Chile, the economic losses due to late spring frosts are in the order of 3 to 7% per year [2]. In the spring of 2013, Chile experienced a severe frost that caused a 22% reduction in exportable fruit, representing a loss of $800 million [3]. The methods for mitigating frost damage are classified into passive and active. Active methods are costly in monetary terms, and some of them have a detrimental effect on the environment, such as diesel heaters. Passive methods are preventive methods, and among them, the use of antifreeze agrochemicals stands out because of their effectiveness and easy application.

Poly(vinyl alcohol) (PVA) is a water-soluble and non-ionic synthetic polymer that contains pendant hydroxyls in the monomeric units. The repeating 1,3 diol units in its backbone chains allow it to be degraded by microorganisms by metabolic oxidation and hydrolysis [4]. In addition, PVA is high biocompatible, and exhibits good film forming ability with oxygen barrier and adhesive properties [5,6]. PVA is typically atactic and is available with varying degrees of hydrolysis [7]. The performance of PVA for a specific application is determined primarily by its molecular weight and degree of hydrolysis. The water solubility of PVA with a high degree of hydrolysis depends on temperature; hydrogen bonds between the hydroxyl groups of PVA chains hinder solubility. In contrast, the solubility of partially hydrolyzed PVA is less temperature dependent; the acetate groups do not contribute to hydrogen bonding [8]. PVA has versatile applications in the pharmaceutical, medical and food industries. Thus, for example, fully hydrolyzed PVAs are used in the formulation of water-resistant adhesive [9], in food packaging [6], and in the development of contact lenses [10]. Soluble PVA macromolecules are used in medical sutures and in the preparation of pharmaceutical capsules for drug release [7]. Recently, we reported the use of an agrochemical formulation based on poly(vinyl alcohol) (PVA) for cherry frost protection [11].

Commercial, polydisperse, atactic and fully hydrolyzed grades (98–99%) of PVAs exhibit similar ice recrystallization inhibition (IRI) activity as structurally different antifreeze glycoproteins [12,13,14]. PVA displays a weak thermal hysteresis [15], and PVA adsorption onto the ice surface does not affect the basal face of ice crystals [16].

The molecular architecture and composition (molecular weight and distribution of pendant -OH and acetate groups) have been evaluated in order to improve the antifreezing properties of PVA [17]. In general, an increase in the molecular weight of PVA leads to an increase in IRI activity. However, the linear, star and bottlebrush architectures of PVAs have comparable IRI activities. Interestingly, the supramolecular structure of PVA homopolymers modified with catechol end-groups and coordinated with Fe^3+^ (PVA: Fe^3+^ ratio of 1:0.33) displayed increased IRI activity [18]. This fact suggests that the volume of the PVA structure plays an important role in the IRI activity.

The mechanism of the IRI activity of PVA is not entirely elucidated, although most studies emphasize the importance of hydrogen bonding between PVA and the ice crystal surface [19,20]. Nevertheless, the adsorption of PVA on ice is weak and reversible. PVA microstructures with a large amount of -OH groups on their surface could increase the IRI activity due to the availability of the -OH groups of PVA to interact with the ice surface.

Few reports on the synthesis of PVA particles have been found in the literature. The preparation of nano/micro-sized PVA particles involves physical [10] and chemical [21] cross-linking through their hydroxyl groups or a combination of both approaches. A chemical method uses crosslinkers, such as glutaraldehyde, or involves irradiation. Zhang et al. [22] reported the preparation of PVA nanospheres, with an average size below 300 nm, by crosslinking in a paraffin microemulsion. The main issue of chemical cross-linking is the presence of chemical residues and their removal from the particle matrix, limiting its use in biomedical and food applications. Cyclic freezing-thawing is a physical cross-linking method for preparing PVA microparticles that involves crystallite formation [23]. In general, this method uses an oil emulsion technique and the average diameter of the PVA nanoparticles is larger than 650 nm. A drawback of this method is the time and solvent consumption of the oil extraction process.

Considering that (i) frost damage to crops is due to the growth of ice crystals in the extracellular space [24], (ii) the PVA macromolecule inhibits the ice nucleation activity of proteins from the ice-nucleating bacterium and the recrystallization of formed ice [25], and (iii) our previous results showed the effectiveness of PVA macromolecules for cherry frost protection [11], here we propose the preparation of PVA microparticles by an alkaline treatment to evaluate their antifreezeing properties. The microparticles were obtained from PVA with high hydrolysis degrees and different molecular weights. Ice recrystallization inhibition was assayed by the splat cooling method and the thermal hysteresis activity of the PVA microparticles by differential scanning calorimetry (DSC). The effect of the PVA microparticles on the susceptibility of flower bud tissues of cherry plants to freeze injury was also evaluated.

## 2. Materials and Methods

### 2.1. Materials

Merck S.A., Santiago, Chile provided three fully hydrolyzed grades of PVAs of different average molecular weights. PVA of Mw = 13,000–23,000 g/mol and 98 % hydrolyzed (SKU: 348406), Mw = 31,000–50,000 and 98–99% hydrolyzed (SKU: 363138) and Mw = 89,000–98,000 and >99% hydrolyzed (SKU: 341584).

### 2.2. Preparation of PVA Microparticles

The PVA microparticles were prepared by alkaline treatment according to the patent N° 201902299 [26]. Briefly, the PVAs were dissolved in nanopure water under magnetic stirring for around 2 h using a mineral oil bath at 100 °C. A 2% *w*/*v* PVA solution was added dropwise to 6% *w*/*v* sodium hydroxide at a flow rate of 30 mL/h using a syringe pump (TE 331, Terumo, Tokyo, Japan) and under constant stirring of 700 rpm. The transparent dispersion was neutralized with hydrochloric acid (37% wt. and density of 1.18 g/mL). Samples were labeled according to the molecular weight of PVA, thus the *ւ*-PVA microparticles correspond to PVA of low Mw; *m*-PVA microparticles, and *h*-PVA microparticles correspond to PVA of medium and high Mw, respectively.

### 2.3. Scanning Electron Microscopy (SEM)

The morphology and size of PVA microparticles before and after the neutralization step were analyzed using a JEOL-JSM 6380LV scanning electron microscope (Tokyo, Japan) operated at 20 kV. One drop of PVA microparticle dispersion was dried at room temperature and coated with a gold film (50 nm). The magnifications used were 2000×, 5000×, and 10,000×. The size of the PVA microparticles was measured directly from the SEM micrographs using the ImageJ software. An analysis of variance to compare the particle size of the samples was carried out and the Tukey’s test was applied (*p* < 0.05).

### 2.4. Fourier Transform Infrared Spectroscopy (FTIR)

The FTIR spectra of PVA microparticles and pristine PVA macromolecules were recorded in triplicate in the range of 4000–600 cm^−1^ at 4 cm^−1^ resolution in a Perkin Elmer Spectrum Two spectrophotometer (Waltham, MA, USA). The FTIR spectrum was taken with attenuated total reflectance (ATR). The background and sample spectra were scanned in transmittance mode. The baseline correction of the spectra was conducted using version 2 of the spectral manager software. According to Pozo et al. [20], the data were preprocessed using Origin version 8.6 software. Pristine PVA macromolecules were also labeled considering their molecular weight. Thus, the *ւ*-PVA macromolecules correspond to PVA of low Mw, *m*-PVA macromolecules, and *h*-PVA macromolecules to medium and high Mw, respectively.

### 2.5. Differential Scanning Calorimetry (DSC)

The thermal hysteresis (TH) of PVA macromolecules and PVA microparticles in the presence of 0.007 M NaCl was determined using a differential scanning calorimeter DSC 6000 (Perkin Elmer Shelton, CT, USA). The equipment was calibrated with indium and nanopure water. The DSC experiments were performed according to references [27,28] with some modifications. The PVA microparticles were dispersed in nanopure water at a concentration of 0.1 mg/mL. A sample of 5 µL was quickly frozen at −30 °C and held for 3 min, followed by heating at 1 °C/min to 15 °C. Then, the sample was cooled to −30 °C again, held for 3 min and heated to partially melt at the same rate. The temperature at which the sample was not completely melted was defined as the hold temperature (T_h_). T_h_ was determined iteratively by the minimization of the amount of ice present at T_h_ which can be estimated from the quotient of phase change enthalpies of freezing and melting. At T_h_, the sample was held for 10 min to allow the PVA microparticle interaction with ice. Then, the sample was cooled again to −30 °C at 1 °C/min. The onset temperature (T_o_) of the crystallization process was recorded. The thermal cycle was repeated at different T_h_ and was carried out under a nitrogen atmosphere (20 mL/min). TH was calculated as the difference between T_h_ and T_o_. The results are the average of three DSC runs of three different samples.

### 2.6. Splat Cooling Assay

The IRI activity was evaluated using a method adapted from Knight et al. [29]. A drop of 10 μL of PVA macromolecule or microparticle samples at 0.1 mg/mL in the presence of 0.007 M NaCl was expelled from a height of 1.85 m onto a microscope slide, previously chilled for 1 h with dry ice. Then, the slide was quickly transferred to the Peltier LTS120 system (Linkam, Salfords, UK) coupled to a BX43 microscope (Olympus, Tokyo, Japan) and held at −6 °C for 1 h. The recrystallization process was followed during annealing at −6 °C using a polarizer for transmitted light (U-POT). Images were captured using an Olympus SC50 video camera at 20× magnification. The area of the ice crystals before (zero hour) and after (1 h) the annealing of the samples at −6 °C was calculated to determine the ability of ice recrystallization inhibition of samples. The difference between the areas of the same ice crystal before and after annealing was calculated using the perimeter of ice crystals measured manually in ImageJ software (74 ice crystals were measured). The area of at least 10 ice crystals per sample was calculated. A 0.007 M NaCl solution was used as a negative control. An analysis of variance to compare the growth rate of the ice crystals among the samples was carried out and Tukey’s test was applied (*p* < 0.05).

### 2.7. Evaluation of PVA in Flower Bud Tissues

During the mid-winter and early spring of 2018, flower buds were taken from 12-year-old ‘Sweetheart’ cherry trees grafted on ‘Colt’ rootstock. Cherry trees were established in a commercial orchard located in San Fabian, Ñuble Region, Chile (36°43′ S; 71°95′ W). Cherry buds from different phenological stages, between bud dormancy (June 19) and bud sprouting (September 12), were carefully opened and placed in containers with cotton embedded in a mineral solution of nitrogen, phosphorus and potassium at 0.09, 0.04 and 0.1% *w*/*v*, respectively. Twenty-four flower buds were sprayed with distilled water (Control), *h*-PVA microparticles dispersion (0.01% *w*/*v*) or *h*-PVA macromolecule solution (0.01% *w*/*v*). Eight flower buds per each treatment and per each phenological stage were used. Sprayed buds were placed in a micro cooling chamber (Calvac, Santiago, Chile) at −5 °C and 70% relative humidity. After 4 h, buds were removed from the microchamber. The freezing damage of flower primordia per bud was evaluated using a stereo microscope SZ61 (Olympus, Tokyo, Japan) connected to a digital camera LC30 (Olympus, Tokyo, Japan). The cumulative flower primordia per bud with freeze damage versus the date of bud collection before flowering was evaluated. The statistical significance of products against freeze injury on cherry buds was evaluated by a non-parametric Kruskal-Wallis test at *p* < 0.05, after normality and homoscedasticity analysis. Infostat software version 2017 (Infostat, Córdoba, Argentina) was used for statistical analysis.

## 3. Results and Discussion

### 3.1. PVA Microparticles Characterization

The alkaline treatment of the aqueous PVA solution produced stable polymer chain aggregates with spherical shapes as shown in Figure 1a,c,e. The size of the PVA microparticles and their size distributions were calculated from at least 190 particles (see Figure 1b,c,f). The average size of PVA particles was 900 ± 300 nm, 1300 ± 400 nm, and 1500 ± 700 nm for *ւ*-PVA, *m*-PVA and *h*-PVA samples, respectively. The mean particle size increased significantly with increasing the PVA molecular weight (Tukey’s test, α = 0.05). The size distributions of the *m*-PVA and *h*-PVA particles were wider than the distribution of the *ւ*-PVA particles. Polymer molecular weight exerts a strong influence on particle size distribution. Legrand et al. [30] evaluated the effect of PLA molecular weight on particle size. The higher the molecular weight, the larger the size of the nanoparticles obtained. This phenomenon was explained by the higher intrinsic viscosity of the precursor macromolecules with high molecular weight. The total volume fraction occupied by the high molecular weight macromolecules is larger than that occupied by low molecular weight macromolecules, so the distance between the chains is shorter and the overlapping of coils is more likely to occur, thus dispersing in larger and smaller polymeric aggregates in the precipitating medium. In general, PVA microparticle samples displayed a high number of associated particles with a necklace shape (see Figure 1). The conformation of PVA in an aqueous solution can explain the effect of the PVA molecular weight on the average size of microparticles. PVA preferably adopts a single chain coiling conformation at 1% *w*/*v* in water where the size of the coil depends on the molecular weight of PVA [8]. However, the interchain distance between the macromolecules of the fully hydrolyzed PVA is short due to the competition between polymer—solvent and polymer—polymer interactions [31]. The addition of the aqueous PVA solution to the NaOH solution decreases the solubility of PVA. The hydration of Na^+^ and OH^−^ ions reduces the number of free water molecules available to interact with PVA. As a result, the polymer—polymer interaction becomes dominant and single coiling chains of PVA tend to aggregate and coalesce, forming coils agglomerates with a spherical shape. Therefore, the size of these micellar structures depends on the polymerization degree of the PVA chain precursor. The PVA microparticles in an alkaline medium showed spherical shapes without aggregation and their size was similar to those described above (see Appendix A). It is proposed that microparticles in the alkaline medium are stabilized by electrostatic repulsion of the negative charge of PVA microparticles. The negative net charge is probably due to the formation of sodium alcoholate on the polymer particles. The FTIR analysis results supported this assumption. The dielectric environment of the PVA microparticles changes by adding HCl, which leads to a pseudo-linear arrangement of microparticles, as observed in SEM (Figure 1a,c,e). The PVA particles maintained their morphology and size after 6 months of storage in an aqueous medium (see Appendix A).

The FTIR spectra of the PVA microparticles and PVA precursors are shown in Figure 2. The spectra revealed the characteristic peaks of PVA [32]. The broad band between 3400–3100 cm^−1^ was assigned to the stretching vibrations of -OH groups and indicated a broad distribution of the chemical environments of these groups. The symmetric and asymmetric vibration of CH_2_ groups was observed at 2910 and 2942 cm^−1^, respectively [22]. The peaks around 1420 and 840 cm^−1^ were associated with in-plane bend and stretch vibrations of CH_2_ groups, respectively. The C-H bending vibrations occurred at 1323 cm^−1^ [33]. The most intense peak at 1090 cm^−1^ corresponded to C-O stretching in aliphatic alcohols [34]. As expected, the vibrations associated with C = O stretching of the acetate group around 1740 cm^−1^ were observed at a lower intensity for PVA precursors with a high hydrolysis degree (98–99%). The band at 1142 cm^−1^, attributed to the C-O stretching vibration, was related to the PVA crystallinity. This band was similar for PVA macromolecules and PVA microparticles [35]. In the FTIR spectra of PVA microparticles, the absence of the peaks between 1750–1735 cm^−1^, corresponding to the C = O and C-O stretching from acetate groups, showed that the alkaline treatment completely hydrolyzed the remaining acetate groups from PVA. This was corroborated by the increasing intensity of the absorbance band at 1568 cm^−1^ in the spectrum of the PVA microparticles, which is associated with the asymmetric stretching vibration of the carboxylate anion of free acetate groups [36]. The adsorption bands of C-H bending shifted from 1323 to 1330 cm^−1^. This band is associated with the formation of alkoxide structures in PVA after alkali treatment [37]. A distinctive feature in the FTIR spectra of the microparticles was the appearance of a new band at 1647 cm^−1^, which could be attributed to the O-H bending of bound water. Generally, this band appeared at a lower wavenumber, around 1635 cm^−1^ for PVA [38], and is largely influenced by the number of hydrogen bonds. The δ(OH) band shifted at a high wavenumber (1647 cm^−1^) indicating that there is a high density of hydrogen bonds in the PVA microparticles available for pairing with water molecules rather than intermolecular hydrogen bonds between particles.

### 3.2. Antifreezing Properties of PVA Microparticles

The splat assay was used to evaluate the IRI activity of PVA microparticles. Figure 3 shows the microphotographs of frozen aqueous PVA macromolecule solutions and PVA microparticle dispersions as a function of annealing time at −6 °C. The PVA macromolecule precursor with NaCl (0.007 M) and NaCl was used as a reference. NaCl in PVA solution and PVA microparticle dispersion ensure liquid water between the ice crystal boundaries [9]. After 60 min of annealing at −6 °C, the PVA macromolecules and microparticles significantly inhibited the ice crystal growth compared to the negative control (0.007 M NaCl, Figure 3m,n). This means that the solution and dispersion showed ice recrystallization inhibition. Furthermore, smaller ice crystals were formed in the PVA microparticles dispersion than in the PVA macromolecule solution. The ice growth rate, expressed as surface area per minute (µm^2^/min), was lower in the presence of PVA microparticles than in the presence of the PVA macromolecule precursor, although the difference among them was not significant, as shown in Figure 3o.

The most accepted mechanism for the IRI activity of PVA proposes that PVA hydroxyl groups precisely and reversibly match the spacing on the prism plane of ice [17]. The authors suggested that ionic species (Na^+^ and Cl^−^) used in the splat assay could reduce PVA solubility in the unfrozen water channels. The ice crystals exclude other solutes and hence concentrate them between the ice grains, leading to the aggregation of PVA macromolecules to form colloids in saline solution [39]. The PVA colloids tend to deposit onto the ice surface, favoring binding between the -OH groups of PVA and the ice. This behavior is highly probable in PVA microparticles due to their large size compared to PVA macromolecules. In addition, PVA microparticles have a high number of free hydroxyl groups to match ice crystals. On the other hand, the effect of the size of PVA microparticles on IRI activity was not elucidated. However, *h*-PVA microparticles showed smaller ice crystals than *ւ*-PVA and *m*-PVA microparticles, suggesting that *h*-PVA microparticles have a higher IRI activity. The increasing IRI activity with the increasing size of the PVA microparticles could be related to the steric hindrance of larger particles adsorbed to the ice surface during ice recrystallization, preventing adjacent ice crystals from coalescing to form larger ones. Larger PVA particles are a more effective physical barrier due to the increased distance between two adjacent ice crystals.

Thermal hysteresis (TH) is the main manifestation of the antifreeze activity of biological or chemical compounds and is defined as the difference between the melting temperature and the non-equilibrium freezing temperature at which the ice crystals start to grow [40]. The temperature of non-equilibrium freezing (T_h_) is usually determined by microscopy observation and is, therefore, observer-dependent. In contrast, in the DSC method, T_h_ is determined by the estimation of the amount of ice present calculated from the heat absorbed and released during phase change. The amount of ice (%) and thermal hysteresis (TH) determined by DSC from PVA macromolecules and microparticles are shown in Table 1. The amount of remaining ice in equilibrium depended on T_h_; with rising T_h_ it decreased from around 20% to 6% and from 20% to 0.3% for PVA macromolecules and microparticles, respectively. The crystallization onset temperature (T_o_) was generally lower for the lower ice fraction at T_h_. This behavior was also reported by Wu et al. [41] when evaluating the thermal hysteresis activity of ice-binding sericin peptides by the DSC method. From Table 1, no significant differences were detected between the thermal hysteresis values of the PVA microparticles or macromolecules and NaCl (TH = 0.3 at T_h_ of −0.2 °C and TH = 0.6, at T_h_ of −0.1 °C). Therefore, PVA macromolecules and microparticles did not present thermal hysteresis. Furthermore, the decrease of T_o_ for PVA macromolecules or PVA microparticles did not depend on molecular weight or particle size. Kristiansen et al. [42] studied the effects produced by the interaction of salts and antifreeze proteins on their TH activity. They proposed that the salt acts by reducing the protein solubility in the solution surrounding the ice crystal. A similar effect could be attributed to NaCl in PVA. The lowered solubility of PVA in the medium by an increasing salt concentration (ice nucleus cannot accommodate salts) causes their shift towards the ice surface region when the ice crystal is partially melted (T_h_).

### 3.3. Evaluation of PVA in Flower Bud Tissues

Climate change, greater thermal fluctuations and warmer springs cause phenological alterations in deciduous fruit crops, such as bud sprouting and flower development [43,44,45]. These sensitive emerging tissues are at greater risk of freeze damage. The challenge for emerging tissues in spring frosts is to prevent or reduce extracellular ice formation. The growth of extracellular ice crystals can cause mechanical damage to plant cells or cellular dehydration [24]. The high IRI activity of *h*-PVA microparticles compared to that of *ւ*-PVA and *m*-PVA microparticles and its PVA macromolecule precursor could be exploited for the freeze protection of sensitive plants, such as cherries, against late spring frosts.

Figure 4 illustrates the effect of the applications of the *h*-PVA microparticle dispersion and *h*-PVA macromolecule solution on the cumulative freezing damage of flower primordia per bud at different phenological stages, between bud dormancy (June 19) and bud sprouting (September 12), exposed to −5.0 °C and 70% of relative humidity for 4 h. Water application was used as a control. In general, the number of flower primordia per bud with freeze damage significantly increased with flower-bud development, confirming the vulnerability of flower organs in a more advanced development stage to freeze damage [43]. The application of *h*-PVA microparticles reduced the injury of flower primordia per bud by around 55% in the tested phenological stages with respect to the control application. This reduction was statistically significant (*p* < 0.05) on June 19 and July 31. However, the application of the *h*-PVA macromolecule solution did not decrease the freeze damage of flower primordia compared to the control, during the early and late rest periods of bud flower development (Figure 4). These results may be related to the morphological aspects of the flower bud tissue and the IRI activity of PVA microparticles and PVA macromolecules. One of the frost protection mechanisms during dormancy in fruit crops is the decreasing of the extracellular water of flower buds during acclimation [46]. The water, macromolecule solution, or microparticle dispersion application improves the frost protection by releasing latent heat as the water freezes (80 cal/g). This protection is effective down to −5 °C for flower buds but is maintained for a short period of time (2–4 min) [47]. After that, a supercooling of the bud occurs, causing lethal damage to reproductive tissue. In the assay of the micro-cooling chamber, the opened flower buds are more susceptible to freeze damage due to non-continuous water application in the absence of effective anti-freezing products. We hypothesize that the application of the aqueous dispersion of PVA microparticles inhibits the growth of ice crystals, i.e., the crystals formed in the opened buds are smaller in size than those formed in the flower buds with water (control) and macromolecule aqueous solution applications. From a physiological point of view, the freeze damage in deciduous fruit trees is related to the water content of the buds [48], which is higher in the surrounding tissues of the growing bud [49]. Thus, during the bud endo-dormancy (deeply rest) the relative water content in cherry buds decreases significantly, whereas near the bud eco-dormancy (late rest) there is a substantial increase in the relative water content [50]. In fact, PVA microparticles seem to effectively inhibit the recrystallization of ice in the surrounding tissues of cherry blossom buds, and therefore, have a high potential to protect against freezing in fruit crops.

The cherry buds were carefully opened for the treatment, i.e., the internal tissue (flower organs) was in contact with PVA during the micro-cooling chamber assay. The penetration ability of PVA microparticles or macromolecules into the bud tissue was not evaluated in this study. During bud dormancy and bud sprouting, cherry flower buds suffer different phenological changes associated with the expansion and development of flower organs, such as pistil, anther filaments, sepals and petals [51]. These changes in size and expansion of the flower organs expose them to freezing damage, but also allow PVA to come in contact with organ flower tissues. Although at the early stage of bud development (June 19), there is less development and expansion of flower organs, these organs are actually differentiated in the bud [51].

## 4. Conclusions

We reported a simple method to prepare PVA microparticles using an alkaline treatment. The average PVA particle size increased significantly with the increase in the molecular weight of the PVA precursor. The PVA particles had observable IRI activity and decreased the freeze damage of cherry flower buds at different phenological stages exposed to low temperatures (−5 °C). This finding shows that PVA particles can be used as an antifreeze agent to prevent freeze damage in fruit crops.

## 5. Patents

There is a patent number 201902299 resulting from the work reported in this manuscript called “Aqueous formulation to reduce frost damage comprising polyvinyl alcohol particles with a molecular weight of 10,000 to 100,000 g/mol and a percentage of acetate groups of 1 to 20%; process for its production and uses of the formulation”.

## Figures and Tables

**Figure 1 polymers-14-02452-f001:**
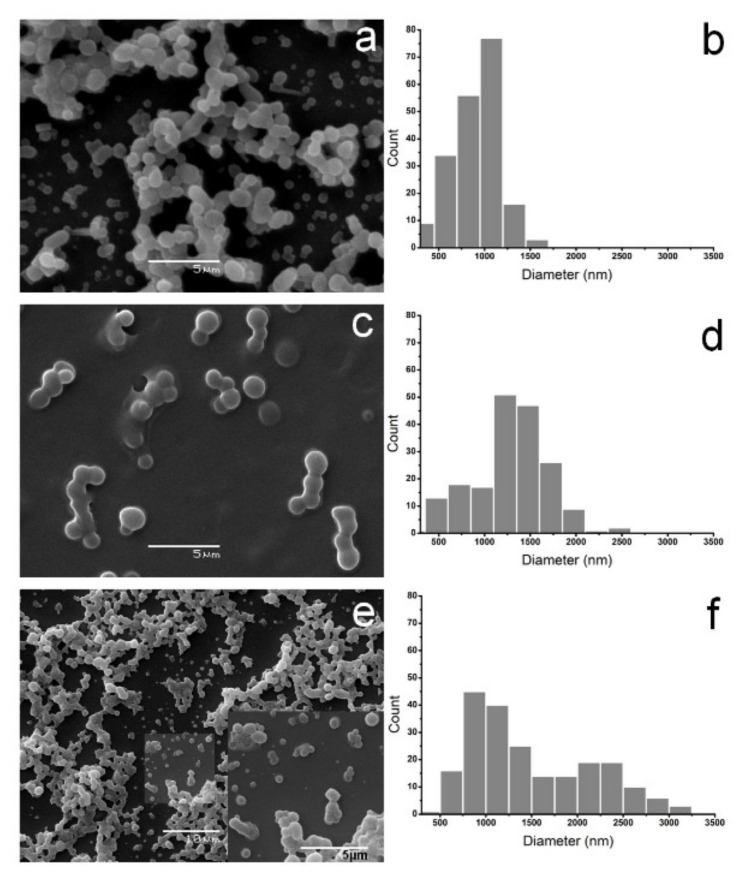
SEM images (**a**,**c**,**e**) of PVA microparticles (*n* ≥ 190) prepared from PVA of different molecular weights and its size distribution histograms (**b**,**d**,**f**). Where: (**a**,**b**) corresponds to *ւ*-PVA microparticles; (**c**,**d**) to *m*-PVA microparticles and (**e**) and f to *h*-PVA microparticles.

**Figure 2 polymers-14-02452-f002:**
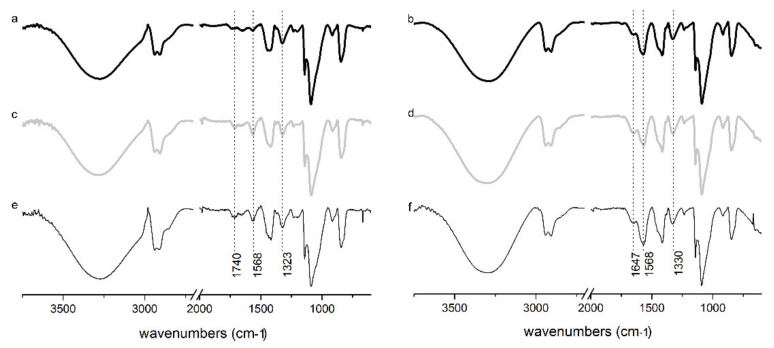
FTIR spectra of PVA of different molecular weights (**a**,**c**,**e**) and PVA microparticles (**b**,**d**,**f**). Where: (**a**,**b**) correspond to *ւ*-PVA; (**c**,**d**) to *m*-PVA and (**e**,**f**) to *h*-PVA.

**Figure 3 polymers-14-02452-f003:**
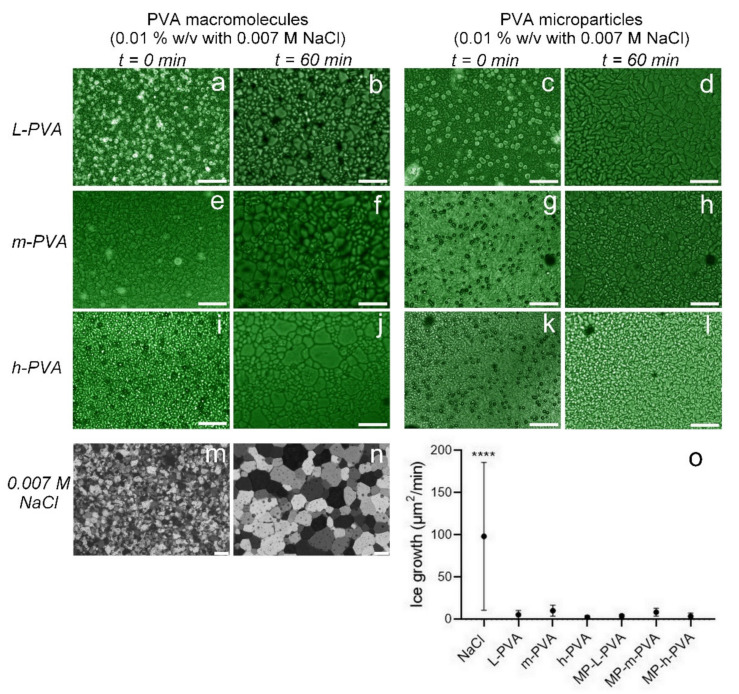
Microphotographs of frozen aqueous PVA solutions (0.01% *w*/*v*) and PVA microparticle dispersions (0.01% *w*/*v*) as a function of annealing time (0 and 60 min) at −6°C. PVA solutions and PVA microparticle dispersions contains 0.007 M NaCl. Where: (**a**,**b**) correspond to *ւ*-PVA solution; (**c**,**d**) to *ւ*-PVA microparticles; (**e**,**f**) to *m*-PVA solution; (**g**,**h**) to *m*-PVA microparticles, (**i**,**j**) to *h*-PVA solution; (**k**,**l**) to *h*-PVA microparticles; (**m**,**n**) to NaCl at 0.007 M. letter (**o**) corresponds to ice growth rate expressed in (µm^2^/min) of the tested samples. Asterisks represent the level of significance among means, *p* < 0.0001 (*n* ≥ 10). Scale bars correspond to 100 µm.

**Figure 4 polymers-14-02452-f004:**
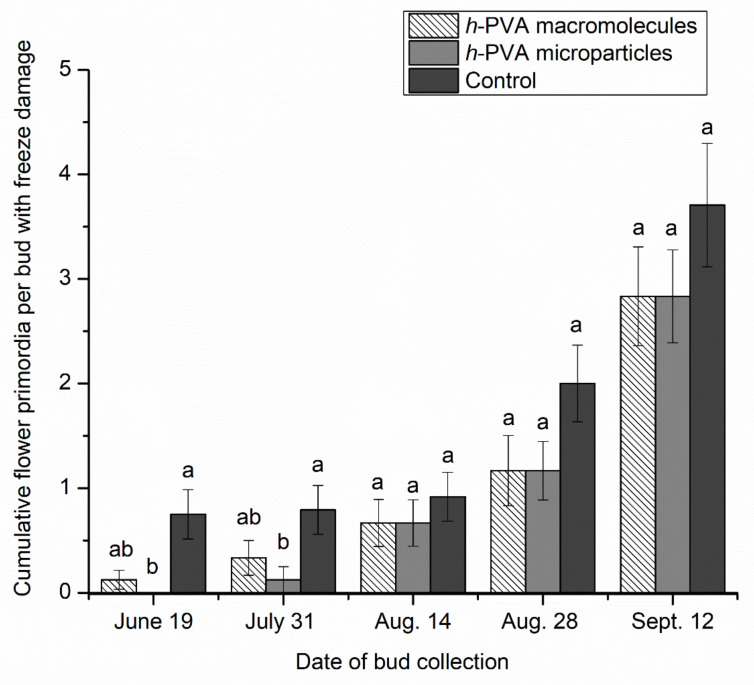
Cumulative freeze damage of flower primordia per bud at different phenological stages between bud dormancy (June 19) and bud sprouting (September 12) exposed to −5.0 °C and 70% of relative humidity for 4 h, after application of water, *h*-PVA microparticles, and macromolecules. Water application was used as a control. Bars with different letters (a and b) indicate a statistical difference at the same phenological stage for *p* < 0.05 by Kruskal-Wallis test.

**Table 1 polymers-14-02452-t001:** Amount of ice (%), onset temperature (T_o_) and thermal hysteresis (TH) determined by DSC of PVA macromolecules and microparticles (0.1 mg/mL), both in the presence of 0.007 M NaCl at different hold temperatures (T_h_: −0.1 and −0.2 °C).

	Amount of Ice (%)		T_o_ (°C)		TH = T_h_ − T_o_	
	Macromolecules	Microparticles	Macromolecules	Microparticles	Macromolecules	Microparticles
T_h_ − 0.2 °C						
*ւ*-PVA	21 ± 2 ^a, •^	s21 ± 1 ^a, •^	−0.54 ± 0.05 ^a, •^	−0.71 ± 0.15 ^a, •^	0.34 ± 0.05 ^a, •^	0.51 ± 0.15 ^a, •^
*m*-PVA	22 ± 5 ^a, •^	20 ± 1 ^a, •^	−0.65 ± 0.02 ^a, •^	−0.6 ± 0.1 ^a, •^	0.45 ± 0.02 ^a, •^	0.4 ± 0.1 ^a, •^
*h*-PVA	19 ± 4 ^a, •^	23 ± 3 ^a, •^	−0.7 ± 0.1 ^a, •^	−0.55 ± 0.06 ^a, •^	0.5 ± 0.1 ^a, •^	0.35 ± 0.06 ^a, •^
T_h_ − 0.1 °C						
*ւ*-PVA	6 ± 3 ^b, •^	0.3 ± 0.1 ^b, ■^	−0.75 ± 0.06 ^a, b, •^	−0.9 ± 0.0 ^a, ■^	0.65 ± 0.06 ^a, b, •^	0.8 ± 0.0 ^b, ■^
*m*-PVA	6 ± 3 ^b, •^	3 ± 2 ^b, •^	−0.7 ± 0.1 ^a, •^	−0.8 ± 0.1 ^a, •^	0.6 ± 0.1 ^a, •^	0.7 ± 0.1 ^b, •^
*h*-PVA	7 ± 1 ^b, •^	3 ± 2 ^b, ■^	−0.7 ± 0.1 ^a, •^	−0.9 ± 0.1 ^a, b, •^	0.64 ± 0.07 ^a, •^	0.8 ± 0.1 ^b, •^

T_h_: hold temperature; T_o_: onset temperature; TH: thermal hysteresis. A 0.007 M NaCl solution was used as a negative control. Values with different letters (a, b) in the same column and T_h_ are statistically different (*p* ≤ 0.05). Values with different symbols (^•^, ^■^) in the same row and T_h_ (comparison between microparticles and macromolecules of same molecular weight) are statistically different (*p* ≤ 0.05). Data were analyzed using *t*-student for two data sets and ANOVA Tukey’s multiple comparisons test (α < 0.05) for three data sets.

## Data Availability

The datasets generated in the current study are available from the corresponding authors on request.

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
