# Peer review of "Preparation of Poly(vinyl Alcohol) Microparticles for Freeze Protection of Sensitive Fruit Crops"

_polymers, 2022, doi:10.3390/polym14122452_

Round 1

Reviewer 1 Report

The results of the research described in this paper can be useful for agriculture to prevent the damage of fruit crops growing in cold environment. In general, the manuscript is well-written and organized.

There are several points which should be improved.

1) Please show the scale bar on the microphotographs in Fig.3.

2) The data in panel O (for ice growth rates) in Fig.3 is not readable because the graph and print are too small. It would be better to place a graph of larger size in separate figure.

3) Section 5. Patents. Since the manuscript is written in English, please provide English translation of the name of the patent number 201902299.

4) It is written in section 2.7 that before the freezing experiments flower buds were sprayed with PVA solutions. This way of treatment assumes that only surface of the buds is covered with PVA cryoprotectant. In this case, it seems only the cells are located near the surface of the buds will be protected from freezing. Does the PVA solution have an ability to penetrate deeply inside bud tissue? Inner tissues in the bulk of the buds are also contain water and can be frozen. It is not clear how PVA on the surface of the buds can also effects growth of ice crystals in the cells far from the bud surface. Please elucidate this issue in Discussion.

Author Response

We are grateful to the Reviewer #1 for its useful comments and suggestions, which have indeed allowed us to improve our manuscript. We appreciate very much the positive evaluation. All the modifications are detailed and explained in the attached file. Please see the attachment. In the revised manuscript, changes have been marked with the Microsoft Word's track changes. 

Reviewer 2 Report

Comments to the Author:

The authors of this manuscript present an interesting research study regarding the preparation of poly(vinyl alcohol) microparticles for freeze protection of sensitive fruit crops. The Introduction and material and methods sections are well described. Also, results are presented in tables and are clear and quite explicable. Due to limited research, I believe that the authors discuss and explain sufficiently the findings of their work. The text needs very few revisions. Although similar research studies are well described in the past the preparation of poly(vinyl alcohol) microparticles for freeze protection of sensitive fruit crop can add further research interest. This is also supported by the patent deposited.

Abstract

COMMENT:

Abstract describes sufficient the findings of this research work.

Introduction

Introduction section is well written and, in my opinion, give the appropriate information without being extended. The purpose of the research work is clearly presented.

Materials and Methods

Please check paragraph style in this section.

Line 123    please delete it

Results and Discussion

Please check paragraph style in this section.

Conclusions

The conclusion section described sufficient the findings of the research work

References

COMMENT:

Although I couldn’t notice any mistake in reference style, I suggest to check reference list once again. Please check capitalised first letter in some words.

In spite of the manuscript is very clear and carefully written some improvement it should be done.

Author Response

We appreciate very much the positive evaluation. The detailed responses are provided in the attached file. Please see the attachment.
